# Repeatability of Brain Activity as Measured by a 32-Channel EEG System during Resistance Exercise in Healthy Young Adults

**DOI:** 10.3390/ijerph20031992

**Published:** 2023-01-21

**Authors:** Christophe Domingos, João Luís Marôco, Marco Miranda, Carlos Silva, Xavier Melo, Carla Borrego

**Affiliations:** 1CIEQV, Escola Superior de Desporto de Rio Maior, Instituto Politécnico de Santarém, Av. Dr. Mário Soares nº 110, 2040-413 Rio Maior, Portugal; 2Exercise and Health Sciences Department, University of Massachusetts Boston, Boston, MA 02125, USA; 3Department of Physics, Instituto Superior Técnico, University of Lisbon, 1749-016 Lisbon, Portugal; 4Department of Bioengineering, LaSEEB-Institute for Systems and Robotics, Instituto Superior Técnico, University of Lisbon, 1049-001 Lisbon, Portugal; 5Centro Interdisciplinar de Estudo da Performance Humana, Faculdade de Motricidade Humana, Universidade de Lisboa, 1496-751 Oeiras, Portugal; 6Centro de Investigação Interdisciplinar Egas Moniz (CiiEM), Egas Moniz School of Health & Science, Caparica, 2829-511 Almada, Portugal

**Keywords:** repeatability, EEG, brain mapping, physical exercise, Emotiv Flex

## Abstract

Electroencephalography (EEG) is attracting increasing attention in the sports and exercise fields, as it provides insights into brain behavior during specific tasks. However, it remains unclear if the promising wireless EEG caps provide reliable results despite the artifacts associated with head movement. The present study aims to evaluate the repeatability of brain activity as measured by a wireless 32-channel EEG system (EMOTIV flex cap) during resistance exercises in 18 apparently healthy but physically inactive young adults (10 men and 8 women). Moderate-intensity leg press exercises are performed with two evaluations with 48 h. between. This intensity allows enough time for data analysis while reducing unnecessary but involuntary head movements. Repeated measurements of EEG during the resistance exercise show high repeatability in all frequency bands, with excellent ICCs (>0.90) and bias close to zero, regardless of sex. These results suggest that a 32-channel wireless EEG system can be used to collect data on controlled resistance exercise tasks performed at moderate intensities. Future studies should replicate these results with a bigger sample size and different resistance exercises and intensities.

## 1. Introduction

Although the relationship between acute aerobic exercise and the cardiovascular system is well-recognized, studies on the impact of acute and long-term exercise on the central nervous system have only recently begun to emerge [1,2,3]. Mounting evidence suggests that the neurochemical changes in the central nervous system induced by exercise-derived metabolites are linked to systematic modulations of brain function [1] that persist after exercise cessation [2]. In this sense, electroencephalography (EEG) has emerged as a feasible application with a high degree of mobility and outstanding temporal resolution [4]. Still, post-exercise EEG has so far provided the only valuable insight into short-term exercise-induced modulations of brain function [2], as EEG data obtained during performance have been limited to ‘closed sports’ (e.g., shooting, archery, etc.), wherein muscle and movement artifacts are not detrimental to the quality of the signals [5]. Recovery EEG data have revealed local phenomena including altered power spectral density [3], alpha peak frequency [6], or oscillatory microstate patterns [7] across exercisers. Importantly, quantifying brain activity response during acute exercise may give valuable mechanistic insight into changes occurring in recovery as it permits the examination of physiological, affective, and perceptual responses [8,9]. However, such brain activity quantification has been limited to aerobic exercise within the moderate-intensity domain and to only a few electrode sites. For example, Kakizaki, 1988 found an increase in the amplitude of the Beta2 frequency range of EEG during cycling at 90 W at a single electrode site (Oz) but found no changes at lower workloads (50–80 W) [10]. Kubitz and Mott, 1996, described changes in EEG during 15 min (three 5 min stages) of cycling with progressively higher workloads (ranging from 50 to approximately 150 W) [11]. During this investigation, electrodes were placed at the frontal (F3 and F4) and temporal (T3 and T4) sites and EEG changes were described globally (i.e., not specific to electrode sites). These authors found no change in alpha or beta activity from before to after exercise but observed reductions in alpha activity and increases in beta activity during exercise. Nybo and Nielsen (2001) described changes in EEG during 1 h of exercise at 60% VO_2max_ in normal (18 °C) and hot (40 °C) environments [8]. No changes were found in EEG, as represented by the percentage change from rest in normal environment (18 °C) in the frontal region (F3), in the midline for the central, and in the midline for the occipital cortices. However, when exercising in a hot environment, increases were observed in the ratio of alpha to beta frequency. Crabbe and Dishman, 2004 conducted a meta-analysis examining EEG responses during and after exercise and concluded that compared to pre-exercise, alpha activity was greater during and after exercise, but there were no differences in relative alpha activity [3]. They also found that the activity of delta, theta, and beta increased during and after exercise.

Resistance exercise may also contribute to the prevention of neurodegenerative diseases, as well as to the maintenance, development, and recovery of brain functions through specific neurochemical adaptations [12,13] that might have different forms of influence on brain activity from those described for aerobic exercise [14].

For EEG to become a useful tool in understanding the impact of various experimental exercise manipulations on brain activity, EEG responses under normal controlled conditions must be repeatable across multiple electrode sites. Highly repeatable measurement of EEG bands is the key to allowing widespread use in longitudinal follow-up studies if exercise training is to be used as a cognitive therapeutic intervention. Surprisingly, the repeatability of EEG bands in exercise-related settings has been overlooked. To our knowledge, only one study examined the repeatability of alpha and beta EEG bands immediately after submaximal running exercise and found them to be good to excellent (ICCs > 0.80) [15]. However, it remains unknown whether this holds during an acute bout of resistance exercise with likely different cortical activities compared to aerobic exercise [14].

Therefore, this study aims to evaluate the repeatability of brain activity as measured by a 32-channel EEG system during resistance exercise in apparently healthy young adults. The benefits of the study are envisioned in three ways. First, a highly repeatable brain activity measurement would suggest its potential use in longitudinal follow-up studies for each subject if resistance training were used as a therapeutic intervention. As a second benefit, a repeatable response to resistance exercise would suggest a highly stable pattern of brain activity. Finally, identifying the areas of the brain that are more active during resistance exercise may pave the way to improve the assessment of brain activity during exercise.

## 2. Materials and Methods

All participants gave written informed consent in accordance with the Declaration of Helsinki [16], after being informed of all the details about the study’s requirements. This study was carried out following the recommendations of local ethics guidelines and was approved by the Ethics Committee of the Sports Science School of Rio Maior, Polytechnic Institute (No3-2022ESDRM) and according to the ethics standards for ethics in sports and exercise science. All data collected were stored in a password-protected database that only researchers related to the brain-mapping project could access. Anonymity was guaranteed by allocating an ID without associating with the person concerned. This research belongs to an EEG brain-mapping project during PE, where the reliability of the EEG device is imperative for further investigation.

### 2.1. Participants

Thirty physically inactive participants (15 men and 15 women) aged 18 to 35 years were recruited to participate in the present study. However, measurements from 12 participants were rejected for noisy readings. Based upon an intraclass correlation coefficient (ICC) estimate of 0.80 from the EEG alpha band repeatability results of a previous study, a power analysis using R package ICC.Sample.Size (version 4.2.1) suggested that the 18 participants left ensured good repeatability in intra- and inter-day repeated measurements (α = 0.05, 1 − β = 0.80, k = 2, null hypothesis = 0.40).

The inclusion criteria were as follows: (1) no history of psychiatric or neurological disorders; (2) no psychotropic medications or addiction drugs; (3) normal or corrected to normal vision, (4) age ranging between 18 and 35 years, and (5) not currently complying with physical activity guidelines. Exclusion criteria included muscle or brain injury in the last 3 months.

### 2.2. Procedures

The evaluations and the intervention program were carried out at the Sports Science School of Rio Maior, IPSantarém and were supervised. The EEG record does not cause any pain or discomfort, and all principles of human health defense were ensured.

The test-retest protocol consisted of two EEG brain mappings on two different days (“Evaluation 1” and “Evaluation 2”), separated by 48 h [17]. It was assured that the same participant replicated the test at the same hour as “Evaluation 1”. The protocol lasted for approximately 50 min.

The EEG head cap was placed before the warm-up to avoid body cooling after the warm-up. Participants were asked to do a slight warm-up of 5 to 10 min consisting of two series of 15 squats, lunges for each leg, lateral squats for each leg, and leg mobility exercises. In addition, subjects were asked to perform 12 repetitions in the leg press so that they could familiarize themselves with the task and the correct motor execution (the starting position was the angular position corresponding to the maximal force production capacity of the knee—70° for the knee extension and 110° for the leg press), while keeping the body warm.

The volunteers performed the 1 RM for the leg press according to the guidelines of the National Strength and Condition Association [18]. First, each individual’s 1 RM was determined: starting with a load corresponding approximately to 50% 1 RM, the participants performed 5 to 10 repetitions. Following this set, the participants performed one set of 3 to 5 repetitions with 60–80% 1 RM. Finally, the participants performed sets of 1 repetition, increasing the load until 1 RM was determined. The rest between repetitions lasted 2 min [19]. All strength tests were accompanied by verbal encouragement. After determining the 1 RM, the loads for 12 RM were calculated (based on approximately 70% of the 1 RM), and volunteers were asked to execute the 12 RM.

### 2.3. Signal Acquisition

There are several electrical brain waves, although researchers still do not fully agree on the exact frequency range and functions of the brain waves. In this study, four of these bands were quantified: the delta band (ranging between 2 to 4 Hz), theta band (ranging between 4 and 8 Hz), alpha band (ranging between 8 and 12 Hz), and beta band (ranging between 12 and 30 Hz) [20]. Of the four, (1) delta is the slowest frequency band with the highest power and associates with sleep states and memory consolidation; (2) the theta band is related to a state of drowsiness, network coordination, and episodic memory; (3) the alpha band is responsible for cognitive performance, selective attention, and consolidation of new motor sequences, and (4) beta is the faster frequency band with the lower power, which relates to sensorimotor association, motor imagery, and autonomous nervous system regulation [21].

The EEG technique is already seen as a robust biomarker for several medical conditions [22], although in the sports domain, due to the environment in which the measures are taken, it is still too early to define reliable biomarkers precisely [21]. Nevertheless, research in clinical [23] and sports [24] domains reported results in relative amplitude, a measure that seeks to present the contribution of oscillations in the different frequency ranges.

During the experiment, the participants performed the physical exercise task in a room where intermittent noise was allowed to maintain the ecology of the task [25]. A 32-channel Epoc Flex system (EPOC Flex Control Box from Emotiv, San Francisco, CA, USA) with various electrode placements within a traditional head cap was used. It was already validated against a Neuroscan device, but for event-related potentials (ERP), changes in alpha signatures, and steady-state visual-evoked potentials [26]. The EEG signals were recorded according to the international 10–20 system (Fp1, Fp2, F7, F3, Fz, F4, F8, T9, C5, C1, C2, C6, T10, T7, C3, Cz, C4, T8, P5, P1, P2, P6, P7, P3, Pz, P4, P8, O9, O1, Oz, O2, O10), with a sampling frequency of 128 Hz. The reference was the average of the left and right mastoids and the signals were recorded by EmotivPro (Emotiv Premium License, San Francisco, USA). During the task, EEG activity was recorded from all 32 channels. The circuit impedance was kept below 5 kΩ for all electrodes before the sessions. Participants were asked to sit comfortably in the leg press, the head remaining as still as possible, and to avoid excessive blinking.

### 2.4. Preprocessing of the EEG

The data were collected in relative amplitude, calculated from 2 to 30 Hz, and then log-transformed. After the signal acquisition, the data was exported to comma separated value (.csv) files, and the name corresponding to each volunteer was codified. Two measures were carried out separated by 48 h and named Eval1 and Eval2. After that, the .CSV files were converted into European Data Format (.edf) to be used in the EDFbrowser software to select the data from the trials for exercise conditions. The collected EEG signals were imported into EEGLAB using the Biosig plugin (pop_biosig.m). For each EDF file, there is a .set file (dataset). The selected data were sent to MATLAB and, more specifically, to EEGLAB to remove artefacts or external noise. The data were high-pass filtered at 2 Hz and low-pass filtered at 30 Hz.

Additionally, some data were rejected using ASR and ICA methods (removal of eye blinking, muscle, heartbeat, or other nonrelated brainwave artifacts); however, at first, the channels with bad signal quality were removed but reconstructed by interpolation from neighboring channels. Relative amplitude values followed the delta (2 to 4 Hz), theta (4 to 8 Hz), alpha (8 to 12 Hz), and beta (12 to 30 Hz) bands. After the data were preprocessed, a more in-depth analysis was possible to extract significant characteristics that could support the experimental objectives. A common approach to distinguishing two different measurements, conditions, or groups is to make a frequency analysis. Different contents of frequency bands can indicate different mental states and, therefore, it is possible to demonstrate some relevant results. Using the FIELDTRIP toolbox, it was possible to create another MATLAB script to compute and present the topographic scalp mapping of APS for Eval1 vs. Eval2 and exercise conditions. This script also allowed the computation and presented the statistical analysis to show the significant differences between Eval1 and Eval2, presenting the topographical scalp mapping of the calculated *p*-values.

### 2.5. Statistical Analysis

The normality distributions of the alpha, beta, theta, and delta EEG bands were tested using the plot representation and the Shapiro–Wilk test. Welch independent-samples *t*-tests were used to test differences in demographics, body composition, and strength outcomes between sexes. Unbiased Hedge’s effect sizes (*g*) for *t*-tests were calculated and interpreted following Cohen’s benchmarks (small (*g* ≤ 0.2), medium (*g* ≥ 0.50), and large (*g* ≥ 0.80)) effects sizes). The repeatability of the EEG bands was examined using parametric ICC (2,1) computed with the irr package and was interpreted as: poor < 0.50, moderate (0.50, 0.74), good (0.75, 0.90), and excellent > 0.90. Additionally, Bland–Altman graphs and statistics were used to assess the repeatability of EEG bands in the 12 RM exercise, using packages ggplot 2. All statistical analyses were conducted using R version 4.2.1 with a significant level set at (α) < 0.05.

## 3. Results

The characteristics of the participants are presented in Table 1. Seven women and five men had incomplete data because of scheduling constraints and were therefore excluded. Regarding the menstrual cycle, 60% of the women were in the follicular phase, ~26.6% in the luteal phase, and the remaining were in the ovulatory phase. Males were taller (*g* = 2.40), had a higher percentage of FFM (*g* = 3.41), and 1 RM in evaluation 1 (*g* = 3.36) and evaluation 2 (*g* = 2.11) compared to females.

### 3.1. Repeatability of EEG Bands during Resistance Exercise

Inter-day repeated measurements demonstrated excellent overall repeatability for all EEG bands during the 12 RM exercise, as the ICCs were >0.90 (Table 2) with a narrow corresponding 95% CI. The repeatability of the EEG bands, when stratified by sex, was also excellent and similar between men and women, although women showed a wider 95% CIs for the ICC point estimate. All EEG bands showed an inter-day bias close to zero considering the total sample and when grouped by sex.

Bland–Altman plots of inter-day repeated measurements suggested no over-proportional bias for the EEG bands during the 12 RM exercise (Figure 1), with the 95% limits of agreement (LOA) relatively wider for the delta band compared to other bands. From visual plot inspection, only one participant out of 18 fell outside the 95% LOA for theta, alpha, and beta, while two for the delta band. Bland Altman plots by sex showed a wider 95% LOA but a bias close to zero for both sexes, and only in females was a proportional bias observed in the theta band.

### 3.2. Topography of EEG Brain Bands under Exercise Conditions

In Figure 2, looking at the different moments, differences were found in the delta band in the P3, PO9, and F4, in the theta band, differences were found in the Oz and T8 channels, and in the alpha band, the FP1 electrode showed differences.

## 4. Discussion

This study aimed to evaluate the repeatability of brain activity measured by a 32-channel wireless EEG system during resistance exercise in apparently healthy young adults who were physically inactive. We found that the EEG measurements were highly repeatable during resistance exercise in all frequency bands, regardless of sex. This suggests that EEG measurements during resistance exercise are a highly stable pattern of brain activity and can be used in longitudinal follow-up studies to track progress or efficacy of resistance-training interventions. The mapping of brain areas and their respective brain waves during resistance exercise also paves the way for improving the assessment of brain activity during exercise in future studies.

The ICCs of the present study support excellent repeatability of EEG bands during resistance exercise (>0.90). Previous studies have reported good-to-excellent repeatability at rest, before or after exercise. For example, Büchel et al. [15] reported a good ICC after participants performed a 10 min run at 50% of their VO2_max_. However, in this study, a 64-channel cap with wet electrodes was used and placed on the heads of the participants after the end of the exercise. Interestingly, EEG data was found to be noisy in eight participants. Similar ICC results were found prior to a cycling exercise task [27]. The ICC estimate was excellent (=0.96), with a 95% CI between 0.722 and 0.997. Although performed at rest, these earlier results are similar to those reported in the present study, wherein EEG measurements were performed during exercise. In 2008, Bailey et al. [28] performed an EEG recording during a bout of cycling exercise and although it was not designed to be a reproducibility study, the results suggested that the EEG could be used to record the brain activity during exercise. However, the authors based their results on only eight electrodes.

The variability of EEG measurements in the present study was found to be higher in the delta and theta bands. These results are not surprising, as a previous study in healthy adults demonstrated that theta, alpha, and beta had strong test-retest reliability compared to the delta band during different doses and types of drugs with central nervous system effects [29]. As expected, the biological behavior of power decreases while the frequency range increases. Interestingly, the results were similar between sex, except for the alpha band, as males presented less variability. This suggests that results from the delta band should be interpreted cautiously or even excluded during resistance exercise, especially when intended to analyze sex differences. This is not the first time women have shown poorer repeatability than men [30,31], probably because of hormonal influences, specifically during the luteal phase [30].

A parsimonious interpretation is that for the 18 participants who were not excluded for noisy electrodes, the repeatability of EEG measurements was excellent, and the equipment appeared efficient in measuring electrical activity during controlled resistance exercises such as the leg press. However, one cannot ignore that noisy readings from 12 participants were discarded. The exclusion of participants with noisy readings is not new and depends on the methodology used in the EEG signal processing, as many researchers choose to smooth and logarithmize the data obtained, instead of excluding it.

Concerning brain mapping, the relative amplitude presents greater activation in the frontal area when looking at the delta band, the central brain area shows more activation in the theta band, the alpha band is more present in the occipital area, and the beta bands are active in the parietal area. All the bands seem to present a normal activation behavior, except for the beta band, which was supposed to show some activation in the temporal area, too. The delta and theta bands showed differences in the inhibition phase, while the alpha band showed differences in the activation phase. Regarding sports tasks, the left prefrontal cortex was found to be associated with implicit motor learning in golf putting tasks [32], and the medial and anterior cingulate areas with focused attention when performing graded exercise on a cycle ergometer [28].

The peripheral electrodes were expected to show differences or higher variability because of their proximity to adjacent muscles or the blinking of the eyes compared to the central electrodes. Greater activation may also be associated with artifacts associated with eye movement that, despite filtering, may still be present.

This study is not without limitations. First, individual variability must be considered because of the stability of the EEG signal. Although the results are positive regardless of the menstrual cycle phase, we did not control for the menstrual phase. It would have been an advantage to have at least 2 min of recording in the resting state immediately before starting the exercise as a comparison factor. Although the ICC and bias were excellent with participants with clean readings, 12 participants were rejected for noisy signals, whereby the study might be underpowered. Intra-individual changes such as familiarization, experimental anxiety, or even subjective well-being were not considered inherent variability sources in the functional connectivity analysis. Therefore, the present study should be considered exploratory.

There is an urge in the sports community to add EEG without considering the reliability or validity of the equipment. However, that should be the first step to ensure the robustness of the research.

## 5. Conclusions

Our findings demonstrate that repeated EEG measurements performed at a moderate intensity during resistance exercise show excellent reliability for scalp-wide profiles. This reflects the stability of the relative amplitude over relatively short time windows (a few days). For the development of diagnostic biomarkers, this reliability is crucial—we would not expect the fundamental biology of the brain to change over several days without intervention, and therefore, biomarkers indexing brain function for diagnostic purposes should not change significantly over this period. However, it is crucial to reinforce that the results must be interpreted carefully since 12 participants were removed because of excessive noise artifacts in the frontal electrodes.

## Figures and Tables

**Figure 1 ijerph-20-01992-f001:**
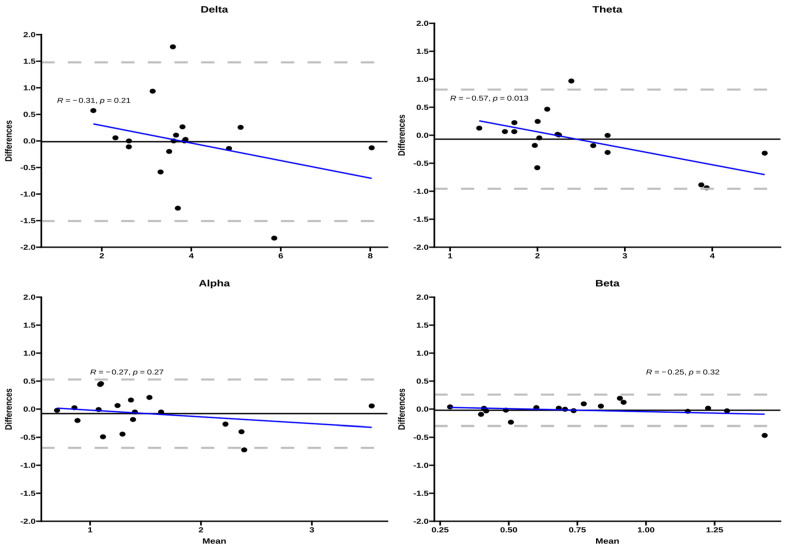
Bland–Altman plots for the EEG bands for the three exercise conditions. The Y-axis corresponds to the difference between inter-day measurements, the dashed gray lines correspond to the 95% LOA, and the black line corresponds to the bias. The blue line corresponds to the regression line between bias and the differences of repeated measures.

**Figure 2 ijerph-20-01992-f002:**
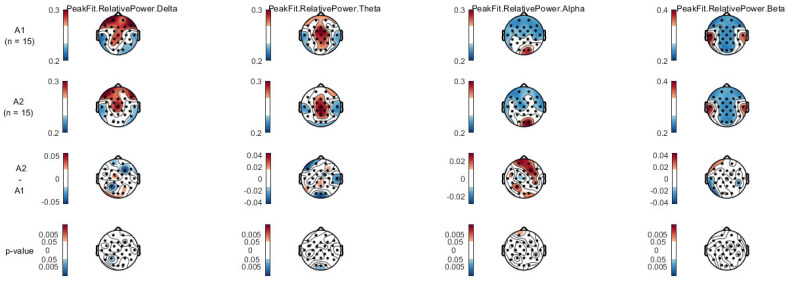
Scale map representation showing the differences between evaluation 1 (first line) and evaluation 2 (second line) for the 12 RM condition.

**Table 1 ijerph-20-01992-t001:** Means (SD) of age, body composition, repetition maximum, and 12 repetition maximum assessed at moment 1 and moment 2.

	All (n = 18)	Female (n = 8)	Male (n = 10)	*p*
Age	21 (3)	20 (2)	21 (4)	0.9
Height (m)	1.70 (0.08)	1.63 (0.05)	1.75 (0.05)	<0.001
Weight (Kg)	66.37 (12.18)	61.05 (15.52)	70.63 (6.88)	0.14
%FM	24.92 (9.88)	31.60 (9.72)	19.57 (6.26)	0.011
%FFM	34.30 (7.10)	27.56 (2.46)	39.69 (4.22)	<0.001
BMI (m/kg^2^)	22.87 (3.43)	22.73 (4.86)	22.97 (1.95)	0.9
1 RM				
Evaluation 1	130.83 (26.64)	105.62 (9.43)	151.00 (15.95)	<0.001
Evaluation 2	130.83 (26.64)	109.38 (18.41)	176.50 (39.16)	<0.001
12 RM				
Delta	3.84 (1.36)	3.91 (1.12)	3.77 (1.59)	0.8
Theta	2.41 (0.77)	2.61 (0.63)	2.26 (0.87)	0.3
Alpha	1.47 (0.68)	1.50 (0.40)	1.45 (0.86)	0.9
Beta	0.75 (0.33)	0.91 (0.29)	0.63 (0.31)	0.078

Abbreviations: M, mean; SD, standard deviation; *p*, *p*-value of Welch *t*-test to compare mean differences of females and males; m, meters; kg, kilogram; FM, fat mass; FFM, fat-free mass; BMI, body mass index; 1 RM, repetition maximal.

**Table 2 ijerph-20-01992-t002:** Inter-day repeatability statistics during 12 RM exercise condition.

	ICC (95%CI)	Bias	SD	95% LOA	
				Lower Bound	Upper Bound
12 RM					
*All (n = 18)*					
Delta	0.93 (0.82–0.98)	−0.01	0.76	−1.51	1.48
Theta	0.94 (0.83–0.98)	−0.07	0.45	−0.96	0.82
Alpha	0.95 (0.87–0.98)	−0.07	0.31	−0.69	0.46
Beta	0.96 (0.89–0.98)	−0.02	0.14	−0.30	0.26
*Females (n = 8)*					
Delta	0.86 (0.22–0.97)	0.06	0.99	−1.98	1.93
Theta	0.87 (0.38–0.98)	−0.14	0.62	−1.34	1.07
Alpha	0.90 (0.54–0.98)	−0.09	0.30	−0.67	0.50
Beta	0.94 (0.68–0.99)	−0.04	0.18	−0.40	0.92
*Males (n = 10)*					
Delta	0.97 (0.89–0.99)	−0.01	0.57	−1.12	1.10
Theta	0.98 (0.91–0.99)	−0.02	0.29	−0.59	0.55
Alpha	0.97 (0.87–0.99)	−0.09	0.33	−0.67	1.09
Beta	0.97 (0.87–0.99)	−0.01	0.11	−0.22	0.21

Inter-day statistics for EEG bands for 12 RM resistance exercise. Abbreviations: ICC: intra-class correlation coefficient; ICC: intra-class correlation coefficient; SD: standard deviations of differences; LOA: limits of agreement.

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
