# Peer review of "Repeatability of Brain Activity as Measured by a 32-Channel EEG System during Resistance Exercise in Healthy Young Adults"

_ijerph, 2023, doi:10.3390/ijerph20031992_

Round 1

Reviewer 1 Report

This manuscript is very well written. The project is scientifically sound. My only recommendation is line 26 in the abstract. "performing two separate evaluation moments of 48 h." is confusing. Suggest something like "performing two separate evaluations with 48 hrs. between."

The main question answered by this research was whether the 32 channel EEG system is a repeatable measure during strength training. This research is important as it is the first step to validating a new type of assessment tool. 

I feel the topic is relevant as a first step. Much more research needs to be conducted to completely validate the EEG system as an assessment of brain activity during exercise. This was pointed out by the authors. Movement during exercise was managed during this study but will be something that needs much more exploration going forward. 

The methodology was sound for what their goal with this project was. Moving forward, looking at ways to mitigate movement during exercise to reduce aberrant measurements is imperative. 

Conclusions are consistent with the evidence. The authors were clear about the limitations of the study and kept their conclusions within the boundaries of the evidence. 

References were appropriate.

My opinion is this is a solid study that is moving research into a new area of utilizing EEGs to measure brain waves during resistance training. This will allow future researchers to have a better understanding of how the brain affects fitness and well being. 

Author Response

We appreciate your kind words and the time you spent reviewing the manuscript.

In addition, we have made the change in line 26.

"Moderate-intensity leg press exercise was performed with two evaluations with 48 hrs. between."

Reviewer 2 Report

Thank you for the opportunity to review the manuscript entitled "Repeatability of brain activity as measured by a 32-channel EEG system during resistance exercise of different intensities in apparently healthy young adults."  Congratulations to the authors for their excellent work. The topic is important for the journal audience, showing that EEG has good to excellent test-retest reliability during a resistance exercise bout. I thought the article was well written, and the methods/statistical analysis are robust. I do not have any major corrections. Listed below are some minor comments.

·         In the title, the authors use the term " different intensities"; however, the research participants just performed the resisted exercise with a 12 RM load. Please review the title of the manuscript.

·         The description of repetition maximum tests and prescription of resistance exercise load is unclear. Why did the authors do a 1RM test? How was the 12RM load estimated from the 1RM test?

·         The authors described the "leg press exercise performed at low intensity." Why do the authors consider a 12RM load as "light intensity"?

·         In Table 1, please describe the measurement unit of the 1RM test.

Author Response

Thank you for your kind words. We appreciate them!

-----------------------------------------------------------------------------------------

You are totally right. We based this title on previous work. The title will be changed accordingly:

"Repeatability of brain activity as measured by a 32-channel EEG system during resistance exercise in healthy young adults.".

-----------------------------------------------------------------------------------------

Thank you for your commentary. To calculate the 12 RM individually, it is necessary to understand the maximal load participants can push (since it is a leg press exercise) first. As mentioned in lines 130-131, we followed the guidelines of the National Strength and Condition Association. 12RM load corresponds to approximately 70% of 1RM. We add more info in lines 140-142.

"After determining the RM, the loads for 12 RM were calculated (based on approximately 70% of the 1RM), and volunteers were asked to execute the 12 RM.".

-----------------------------------------------------------------------------------------

Once again, you are right, and changes will be made in accordance:

"leg press exercise performed at "moderate intensity." – Line 27 and Lines 348-349

-----------------------------------------------------------------------------------------

Changes were made accordantly. 

Reviewer 3 Report

This study aimed to understand whether the EMOTIV flex cap with 32 channels shows repeatability and can be used in physical exercise tasks. From the sports and exercise fields' point of view, this study could bring interesting output to understand the connection between the brain and exercise. However, it is believed that using these technologies in sports situations where decision-making is relevant is the next challenge step point. Some points may be better explained in the text. 

  1. Power analysis estimations may be placed in the participants' section. Nevertheless, it is unclear how the procedure was about sample Size. First, the authors said that thirty physically inactive participants (15 men and 15 women) aged 18 to 35 years were recruited to participate in the study. Then, Size (2) suggested a minimum of 18 participants to ensure good re-204 repeatability in intra and inter-day repeated measurements (α= 0.05, 1−  =0.80, k =2, null 205 hypotheses = 0.40). If they only needed 18 participants for the experiment, why is the collected data in 30? Why didn't they use the complete data available? 
  2. Two separate evaluation moments of 48 h. is there any rationale about this frame time between assessments? 
  3. Details about the sample and selection are needed. 
  4. The rest between repetitions was of 2 minutes. Was it kept like this in all situations? Did the authors follow any specific protocol? References are needed.
  5. How long does the experiment with each person? 
  6. Is there more than one EEG head cap in this experience? Did the participants use the same EEG head cap in moments one and two?
  7. Why was it considered the inclusion criteria "not comply with the recommendations for physical activity". Does the author think it will be different in a more active population? 
  8. Do the authors expect similar patterns in active athletes?
  9. The authors may want to discuss in more detail the conceptual and applied implications of the findings.
  10. Additional factors that may modify the relations observed need to be considered in more detail in the discussion. 
  11. Is there any particular reason to use the equipment "A 32-channel Epoc Flex system (EPOC Flex Control Box from Emotiv, San Francisco, USA)"? is there any previous validation study published by the brand that supports the results achieved with the equipment? 
  12. Which Statistical analyses were performed to compare groups in table 1. 
  13. The major concerns about this study were well described in the limitations section. I assume that a study with such limitations could introduce strong bias in the interpretations of the results. 
  14. The second major concern is what this study adds. What is the main contribution? 

Author Response

Power analysis estimations may be placed in the participants' section. Nevertheless, it is unclear how the procedure was about sample Size. First, the authors said that thirty physically inactive participants (15 men and 15 women) aged 18 to 35 years were recruited to participate in the study. Then, Size (2) suggested a minimum of 18 participants to ensure good re-204 repeatability in intra and inter-day repeated measurements (α= 0.05, 1− b =0.80, k =2, null 205 hypotheses = 0.40). If they only needed 18 participants for the experiment, why is the collected data in 30? Why didn't they use the complete data available?

Answer: Thank you for your question. This is a key question, and if the reviewer has questions about it, then it needs to be explained properly.

It all starts with the way researchers analyze the data. There is no guideline, and there are researchers applying light filters and others who are more conservative. In our case, we chose to be more transparent with our results. We were conservative to the point of excluding electrodes (the alternative often used is to extrapolate values for these electrodes).

Therefore, 30 people were effectively recruited, but after filtering the data, 12 participants had too much noise (especially in the frontal area) to be considered eligible regarding our conservative data analysis. However, the repeatability power analysis was adjusted to consider the participants excluded due to noisy electrodes (α= 0.05, 1− b =0.80, k =2, null 205 hypotheses = 0.40).

"Thirty physically inactive participants (15 men and 15 women) aged 18 to 35 years were recruited to participate in the present study. However, measurements from 12 participants were rejected for noisy readings. Based upon an intraclass correlation coefficient (ICC) estimate of 0.80 from the EEG alpha band repeatability results of a previous study (1), a power analysis using R package ICC.Sample.Size (2) suggested that the 18 participants left ensured good repeatability in intra and inter-day repeated measurements (α= 0.05, 1− b =0.80, k =2, null hypothesis = 0.40)."

We will add to the limitation that the study might be underpowered, and we will reinforce in the conclusion that although the repeatability for the 18 subjects was excellent, 12 subjects had to be excluded due to electrode noise, so the equipment must be used with caution and the results interpreted carefully.

"Although the ICC and bias were excellent with participants with clean readings, 12 participants were rejected for noisy signals, whereby the study might be underpowered. "

"However, it is crucial to reinforce that the results must be interpreted carefully since 12 participants were removed due to excessive noise artifacts in the frontal electrodes."

Two separate evaluation moments of 48 h. is there any rationale about this frame time between assessments?

Answer: This is a good question, and we will add some references in the text to support it. Indeed, it is known that some markers such as ammonia, growth hormone, and creatine kinase suggest a recovery up to 24–48 h post-exercise (Morán-Navarro, R., Pérez, C.E., Mora-Rodríguez, R. et al. Time course of recovery following resistance training leading or not to failure. Eur J Appl Physiol 117, 2387–2399 (2017). https://doi.org/10.1007/s00421-017-3725-7)

Details about the sample and selection are needed.

Answer: This data will be added to question 1.

The rest between repetitions was of 2 minutes. Was it kept like this in all situations? Did the authors follow any specific protocol? References are needed.

Answer: Reference will be added.

Hulmi, J.J., Kovanen, V., Selänne, H. et al. Acute and long-term effects of resistance exercise with or without protein ingestion on muscle hypertrophy and gene expression. Amino Acids 37, 297–308 (2009). https://doi.org/10.1007/s00726-008-0150-6

How long does the experiment with each person? 

Answer: We appreciate your question. We will add that the protocol lasted for approximately 50 min.

Is there more than one EEG head cap in this experience? Did the participants use the same EEG head cap in moments one and two?

Answer: The EEG cap is always the same to avoid even more measurement limitations.

Why was it considered the inclusion criteria "not comply with the recommendations for physical activity". Does the author think it will be different in a more active population?

Answer: Thank you for your interesting question. The plasticity of the brain is different between physically active and physically inactive adults at a cognitive level (Domingos et al., 2020). Based on that previous work, the authors of this study believe that changes can be found between these groups, and future studies will replicate the methodology of this study but in physically active subjects.

Do the authors expect similar patterns in active athletes?

Answer: It was answered above.

The authors may want to discuss in more detail the conceptual and applied implications of the findings.

Answer: As mentioned in the first question, more information will be added to the manuscript.

Additional factors that may modify the relations observed need to be considered in more detail in the discussion.

Answer: It was answered above.

Is there any particular reason to use the equipment "A 32-channel Epoc Flex system (EPOC Flex Control Box from Emotiv, San Francisco, USA)"? is there any previous validation study published by the brand that supports the results achieved with the equipment?

Answer: One study validates the hardware but considers ERP (William, et al., 2020), not the power spectrum. We are the first to do so during exercise. That is why we want to bring this insight to researchers.

Which Statistical analyses were performed to compare groups in table 1.

Answer: Welch independent-samples t-tests were used to test differences between sex for participants' characteristics depicted in table 1. This information was added to the statistical analysis section of the manuscript

The major concerns about this study were well described in the limitations section. I assume that a study with such limitations could introduce strong bias in the interpretations of the results.

Answer: We totally agree with your sentence. Moreover, this is why researcher needs to understand that advances in the EEG domain exist. However, it is still premature to assume some published results.

The second major concern is what this study adds. What is the main contribution?

Answer: Already mentioned previously. We will make sure to reinforce our conclusion.

Reviewer 4 Report

I enjoyed reading this article very much. It is a novel topic and the experiment was straightforward and well-presented, with interesting and applicable conclusions.

My knowledge of EEG is minimal but I feel as though I was able to follow along with the methodology. Hopefully, other reviewers can address those specifics.

Only minor grammatical changes are suggested.

Abstract

No comments

Introduction

No comments

Methods

Line 113 “age comprehended between..” 

I am not sure what this means

Line 114 “not comply with the recommendations for physical activity

Suggest: not currently complying with physical activity guidelines or  not currently complying with physical activity recommendations

Line 123  EEG cap was placed on the heads—did you also connect electrodes at that point? This question is from a methodology repeatability standpoint

Line 128 – move the degree symbol one space closer to the number to avoid the degree symbol on its own

Line 131 “First, RM was determined: …”

Suggest: First, each individual's 1RM was determined:…

Also suggest keeping a similar form throughout either 1-RM or 1RM

Line 135 “The rest between repetitions was of 2 minutes.”

Suggest: The rest between repetitions was 2 minutes

Line 137 change RM to 1RM

Line 151 “This technique is already seen as a robust biomarker…”

I am not sure which technique?— measuring the 4 brain waves?

And what are you biomarking? May need additional explanation.

Line 156 “…performed the PE task…”. Please define PE

Lien 188 Is this the start of a new paragraph?

Discussion

Line 315 “However, it cannot be ignored that 12 participants suffered, in at least one moment, that caused the invalidity of the recorded data.”

I think this is misstated- did the participants themselves suffer or did their data suffer from excessive artifact? A language glitch

Line 316 “As previously mentioned, the exclusion of participants, for this reason,..” Related to the above --  participants had discomfort in the process—please clarify why they would be excluded.

Author Response

Thank you for the time you spent reviewing the manuscript and the honesty of your commentaries. We are sure that your suggestions will improve our work.

-----------------------------------------------------------------------------------------We changed to:

"age ranging between 18 and 35 years old.". – Line 116.

-----------------------------------------------------------------------------------------

Changes made accordingly.

"not currently complying with physical activity guidelines.". – Lines 116-117.

-----------------------------------------------------------------------------------------

Thank you for your question. This cap already has the electrodes connected.

-----------------------------------------------------------------------------------------

Changes were made accordingly in line 132.

-----------------------------------------------------------------------------------------

Thank you for your suggestion. Changes were made in all the manuscript regarding this matter.

-----------------------------------------------------------------------------------------

Changes made accordingly.

-----------------------------------------------------------------------------------------

Already done previously.

-----------------------------------------------------------------------------------------

Thank you for your questions. Indeed, we need to add that we are talking about the EEG technique. Moreover, the EEG is used as a biomarker (brain map and signal processing) to detect many brain pathologies in their early stages. And that's the point of EEG in sports. We don't have any way to use EEG as a biomarker yet, so we need to start collecting data during exercise.

-----------------------------------------------------------------------------------------

Already done, thank you.

-----------------------------------------------------------------------------------------

Some modifications were made to the body text, so we are not sure what line you are referring to. We are sorry.

-----------------------------------------------------------------------------------------

Thank you for noticing that glitch. We made some modifications:

"However, one cannot ignore that noisy readings from 12 participants were discarded. The exclusion of participants with noisy readings is not new and depends on the methodology used in the EEG signal processing, as many researchers choose to smooth and logarithmize the data obtained instead of excluding it." – Lines 316-319.

Round 2

Reviewer 3 Report

This is a nice paper on  Electroencephalography, namely,  the impact of acute and long-term exercise on the central nervous system. 

The questions raised in the first phase were satisfactorily answered. We don't have further comments on the paper.